# Depth Range Extension for the Misty Grouper *Hyporthodus mystacinus* Documented via Deep-Sea Landers throughout the Greater Caribbean

Shannon E. Aldridge [1,*], Olivia F. L. Dixon [1], Christine de Silva [1], Johanna K. Kohler [2], Oliver N. Shipley [1], Brennan T. Phillips [3], Teresa F. Fernandes [4], Timothy Austin [2], Rupert F. Ormond [4,5], Mauvis A. Gore [4,5] and Austin J. Gallagher [1]

1   Beneath the Waves, 3 Austin Street, P.O. Box 290036, Boston, MA 02129, USA;
    liv@beneaththewaves.org (O.F.L.D.); christine@beneaththewaves.org (C.d.S.);
    ollieshipley7@gmail.com (O.N.S.); austin@beneaththewaves.org (A.J.G.)
2   Cayman Islands Department of Environment, 580 North Sound Road, P.O. Box 10202, George Town KY1-1002,
    Cayman Islands; johanna.k.kohler@gmail.com (J.K.K.); timothy.austin@gov.ky (T.A.)
3   Department of Ocean Engineering, University of Rhode Island, 15 Receiving Road, Narragansett, RI 02881,
    USA; brennanphillips@uri.edu
4   Institute of Life and Earth Sciences, Heriot-Watt University, Edinburgh EH14 4AS, UK;
    t.fernandes@hw.ac.uk (T.F.F.); rupert.ormond.mci@gmail.com (R.F.O.); mauvis.gore.mci@gmail.com (M.A.G.)
5   Marine Conservation International, South Queensferry, Edinburgh EH30 9WN, UK
*   Correspondence: shannon@beneaththewaves.org

**Abstract:** Misty Groupers (*Hyporthodus mystacinus*) are one of the largest and most geographically widespread grouper species and one of the few grouper species known to occur at depths greater than 200 m. However, aspects of their basic biology, behavior, and ecology remain poorly understood, leaving significant gaps in our ability to evaluate their functional role throughout the vertical water column, as well as our understanding of their conservation needs in a changing ocean. Through in-situ video observation obtained using deep-sea landers in both The Bahamas and Cayman Islands over multiple years, we documented Misty Grouper occurrence up to 470 m depth in the mesopelagic zone. These observations provide a new depth range extension for the species and illuminate the potential importance of deep-water habitats for large grouper species in the wider Caribbean.

**Keywords:** Misty Grouper; deep-sea; MPA; Caribbean; depth; BRUV

**Key Contribution:** This report documents the presence of Misty Groupers via deep-sea lander videos at depths up to 470 m, extending the previously known depth range for the species. Additionally, the overall study emphasizes the need to better understand deep-sea connectivity, diversity, and importance in today's changing oceans.

## 1. Introduction

Mesopelagic fishes that commonly inhabit waters ranging from 200–1000 m deep also often spend time along the bathyal zone (1000–4000 m) and play an important role in the distribution of organic matter throughout the water column [1]. These fishes are capable of feeding and excreting in both the upper water column and at depth, providing trophic connectivity and potentially influencing biogeochemical cycles [1]. Understanding the connectivity between marine zones would allow for better planning of MPAs and also allow for a better understanding of how species react to disruptions, such as changing ocean circulation patterns, climate change, and overfishing [2]. While the technological advances made within the last few decades are significant, the ability to deploy deep-sea research units can be challenging due to the high cost of appropriate marine vessels [2] and the challenges that come with deep-sea research, such as pressure-tolerant equipment,

adequate battery power, no natural light, sometimes unknown bathymetry, and significant amounts of fuel required to reach drop locations [3]. A method that is now becoming more common for deep-sea research due to being both a non-invasive technique and more affordable compared to traditional long-line fishing is Deep-sea Baited Remote Underwater Video systems (hereafter dBRUV) [4].

Groupers (family Serranidae) can be found worldwide and are significantly diverse when it comes to size, coloration, and habitat preference. It is generally understood that groupers, particularly large groupers, utilize shelves and walls for activities such as foraging and spawning. Misty Groupers (*Hyporthodus mystacinus*), Poey, 1852, can be found throughout the western North Atlantic, the Caribbean, and the eastern Pacific around the Galapagos Islands [5] and are one of the largest grouper species, with a maximum recorded weight of 107 kg and a total length of 160 cm according to a reference cited in Fishbase [6], though it has been assumed in field guides that they can grow larger. Though there have been very few studies conducted for this species, it is commonly thought that Misty Groupers are specialized deep-water predators [7]. It has been assumed that they are protogynous hermaphrodites that exhibit very slow-growing characteristics [5].

Groupers are important to both recreational and commercial fisheries throughout the Atlantic, particularly the Misty Grouper for deep-sea commercial fisheries. Fisheries typically move to a deep-water approach when resources nearshore become depleted due to events such as overfishing or habitat loss [8]. Deep-water fishing has also become more readily practiced with the advancement of the necessary technology, such as depth sounders and more accurate GPS [8]. Misty Groupers have been documented as the most dominant grouper caught by the US Virgin Islands deep-water fishery [5] and are a significant contribution to the Puerto Rican fishery [9]. Despite the species' economic importance, there is very little known about Misty Grouper biology and ecology. Additionally, studies that have involved surveys of the species thus far have been conducted utilizing invasive long-line catch techniques. Finally, studies involving Misty Groupers have been conducted in Bermuda [9], Navassa Island [10], Mesoamerica [11], and the Galapagos Islands [12], but neither The Bahamas nor the Cayman Islands. While it has been assumed in field guides that Misty Groupers can be found at up to 400 m in depth, the deepest recorded depth of a Misty Grouper in primary scientific literature prior to this report was 272.1 m off the coast of Navassa Island in the Caribbean Sea [10]. A noteworthy observation from the National Oceanic and Atmospheric Administration during the Okeanos Explorer ROV Dive on 15 November 2018 was the occurrence of a Misty Grouper at the beginning and towards the end of the dive, and while the report does not indicate specifically what depth the fish was observed, the maximum depth of the dive was 366 m [13].

Here, we report two sightings of Misty Groupers, one in The Bahamas Islands and one in the Cayman Islands, at depths greater than previously recorded in the primary scientific literature for the species via deployment of deep-sea landers equipped with video cameras. This information provides evidence to expand the known depth range of the species and provides reason for considering greater protection of deep-sea habitats.

## 2. Materials and Methods

The Bahamas archipelago is made up of approximately 3000 carbonate islands, rocks, and cays located in the western North Atlantic that developed approximately 200 million years ago while the Atlantic Ocean was forming. There are two primary deep-water channels that split the major banks: the Tongue of the Ocean (hereafter TOTO) and the Exuma Sound. Both are V-cut canyons at the bottom of a U-shaped trough, but the TOTO extends greater than 4000 m, and the Exuma Sound maximum depth is assumed to be approximately 1600 m [14].

The Cayman Islands are part of the southern portion of the North American Plate known as the Cayman Ridge and consist of three islands: Grand Cayman, Little Cayman, and Cayman Brac [15]. The islands were formed approximately 10 million years ago via block faulting and uplifting [16], and there is a unique oceanographic and marine geological

feature in direct proximity to the islands: the Cayman Trench. This trench is over 7000 m in depth and is located at the edge of the North American plate between it and the Caribbean plate south of the trench [16]. Most of the studies surrounding the fishes of the Cayman Islands reference Burgess' 1978 unpublished thesis [17].

Considering the potentially significant biodiversity and insights of the deep-sea, The Bahamas and the Cayman Islands should be considered for continued research [4,18]. As part of two ongoing four-year studies (2020–2023) to survey the biodiversity of large sharks and fishes in the territorial deep-waters of The Bahamas Islands and the Cayman Islands, surveys were conducted using a custom free-falling dBRUV lander system [4,18].

The dBRUV system was used in both regions of study, and assembly instructions were provided by Gallagher et al., 2023 [18] (Figure 1). The primary structure of the dBRUV was a carbon-fiber frame with a pressure-tolerant flotation (G2200, McLane Research Laboratories Inc., East Falmouth, MA, USA), which results in a vertical orientation, and an acoustic weight-release system (CAT, EdgeTech, West Wareham, MA, USA). A depth-rated radio transmitter was attached to the frame (F1845B, Advanced Telemetry Systems, Isanti, MI, USA), and an orange identification flag was attached to the flotation to facilitate recovery at the surface. A single GoPro camera (either GoPro Hero 5, 6, or 7, since all have similar capabilities, and with the nature of field work, there were occasionally last-minute equipment switches made), set to record 1080p video at 60 frames per second continuously until the unit was retrieved or the external battery exhausted, was secured within a deep-water housing (GoBenthic, GroupB Incorporated, Jensen Beach, FL, USA) and attached approximately 1.5 m above the bottom of the unit on the carbon-fiber frame. Two LED lights with an output of 480 lumen each and colors 3000, 4240, and 6020 were used to illuminate the seafloor (ICELite, Juice Robotics, Middletown, RI, USA). A custom lithium-ion battery pack provided power for the camera and lights. Temperature and depth were recorded using either a calibrated Starmon TD stand-alone logger (Star Oddi, Gardabaer, Iceland) or LAT1400 temperature and depth logger (LOTEK, Seattle, WA, USA). Approximately 500 g of *Sarda* spp. were placed in a bait cage that was mounted on a pole in front of the camera. A generic bait fish that could be easily obtained was used for all dBRUV deployments throughout both studies in order to remain consistent. Upon command with an acoustic release system (PORT LF-SD, EdgeTech, West Wareham, MA, USA), the drop weight (sacrificial cinderblock) was released, allowing for the entire system to return to the surface, resulting in deployments of 2–8 h throughout the entire study. All dBRUVs were then located and retrieved at the surface using a combination of boat-based GPS unit, YAGI antenna, and VHF radio receiver (R410, Advanced Telemetry Systems, Isanti, MI, USA), depending on whether the unit was visually spotted or required additional equipment to locate. Upon retrieval of the dBRUVs, videos were downloaded from the camera's micro-SD card and reviewed by a person at $0.5\times$ regular speed. Due to the fact that it was a multi-year study, publications thus far have been unique sightings prior to an overall report that will be published at a later date. All research was conducted under scientific research permits acquired from The Bahamas Department of Marine Resources (DMR) and the Cayman Islands Department of Environment (DOE).

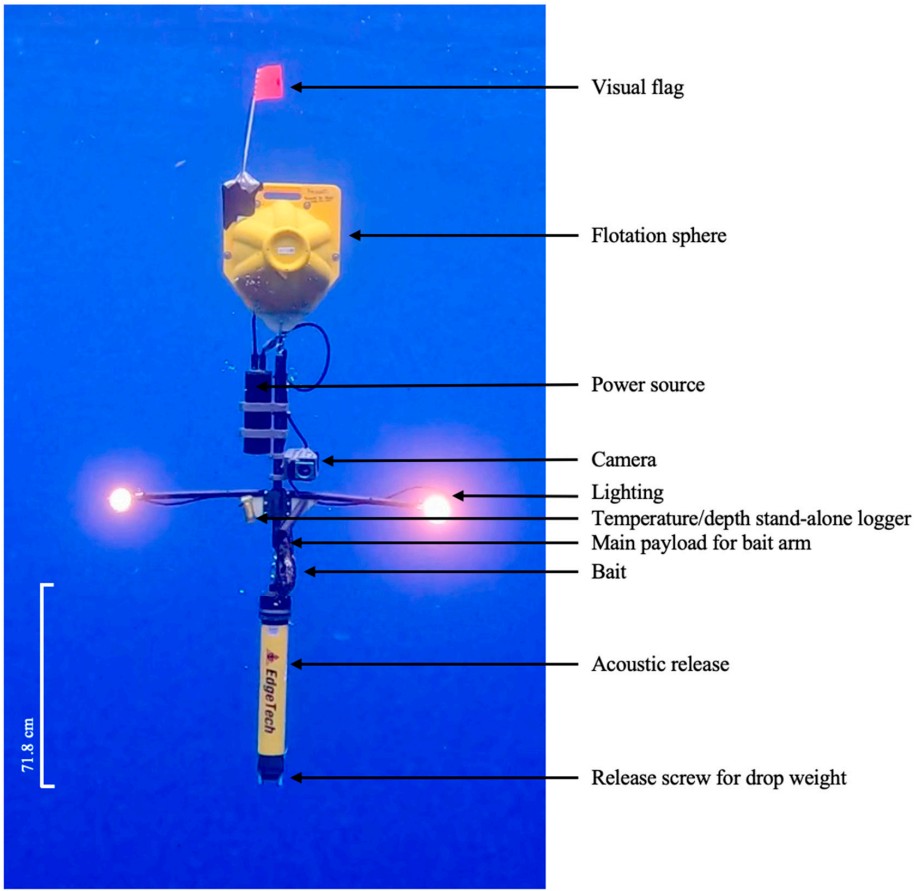

**Figure 1.** Diagram of the deep-sea BRUV (dBRUV) video camera/lander system used in the present studies, with associated components. Filmed during ascent after the sacrificial weight was released.

## 3. Results and Discussion

On 15 January 2020, a dBRUV was deployed 1.0 km off the southeast side of the Berry Islands, Bahamas (25.391950° N, −77.777433° W), a northern point of the TOTO, at a depth of 470 m and recorded 7 h and 55 min of video overnight between 20:00 and retrieval the following day (Figure 2A). A Misty Grouper, identified by its evenly-spaced, dark vertical banding and distinct dorsal spine morphology, was first observed 25 min after deployment; it was swimming within view of the camera as the dBRUV unit was rotating due to a recent landing on the benthos (Supplementary Video S1). The Misty Grouper was observed on camera for a total time of 13 s (Figure 3A). It was assumed that, based on the location of the individual and the dBRUV rotating, the same individual was observed in the following detection, during which the camera rotation significantly slowed due to it settling into position. The Misty Grouper was observed slowly swimming on the edge of the detection window (i.e., the illuminated seafloor in the field of view), hovering in one location at times. The individual remained on camera for a total of 97 s before swimming out of view. At 37 min after deployment, a Cuban Dogfish (*Squalus cubensis*) interacted with the bait and caused the dBRUV unit to slowly rotate again, with a subsequent Misty Grouper observation. The fish was observed swimming slowly and occasionally hovering above the seafloor for 37 s before the dBRUV unit rotated into a position where the fish was no longer in view. It could not be confirmed whether all grouper detections were of the same individual. For this overall survey, the species was observed within the detection window for a total of 147 s, and only one individual Misty Grouper being observed at a time. The habitat observed for this drop site consisted primarily of coarse sand (0.25–2 mm) with a shallow slope and low rugosity. There were occasional rock features observed during unit rotation.

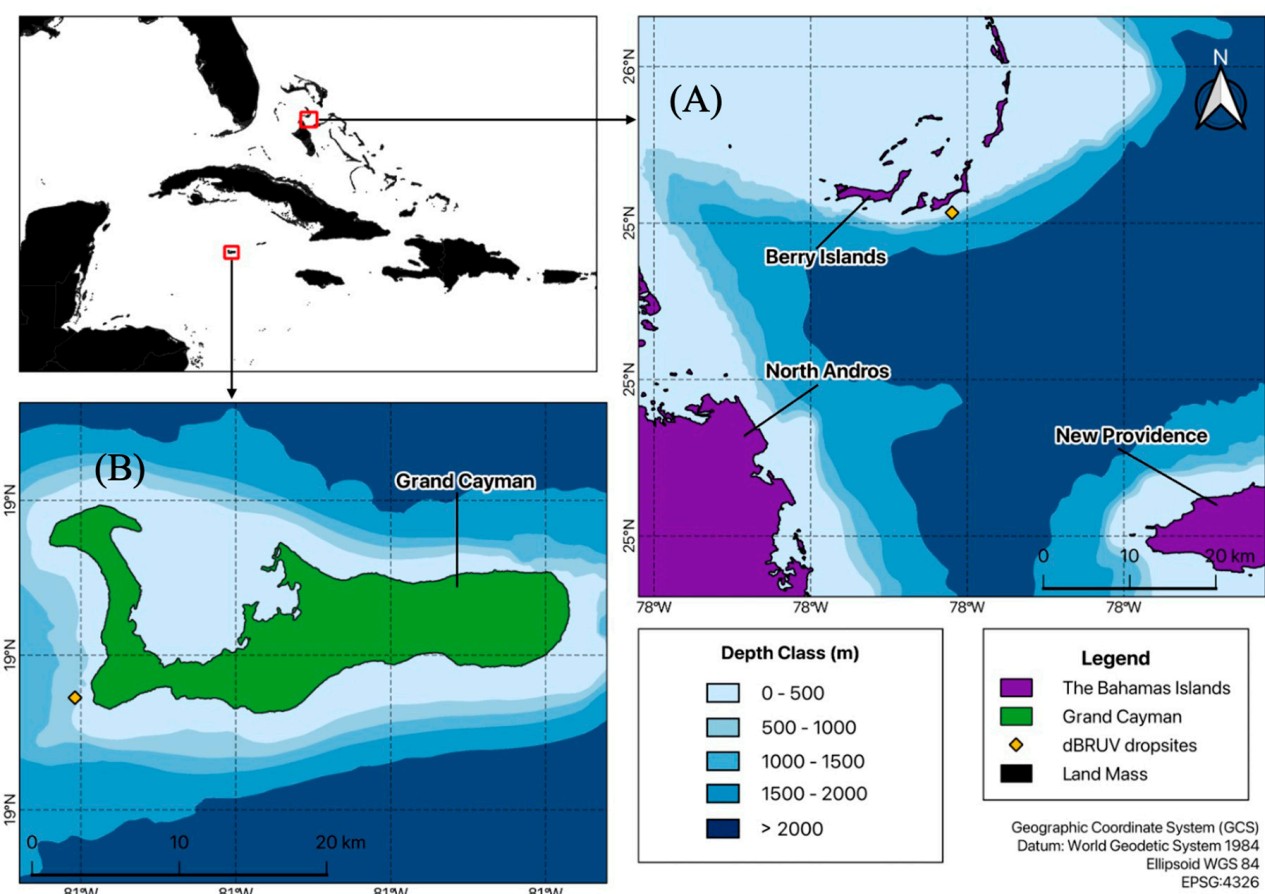

**Figure 2.** Deployment locations of dBRUV units. (**A**) The Bahamas Islands, 15 January 2020. (**B**) Cayman Islands, 7 April 2023. The yellow diamonds indicate the drop sites where Misty Grouper occurrences were observed.

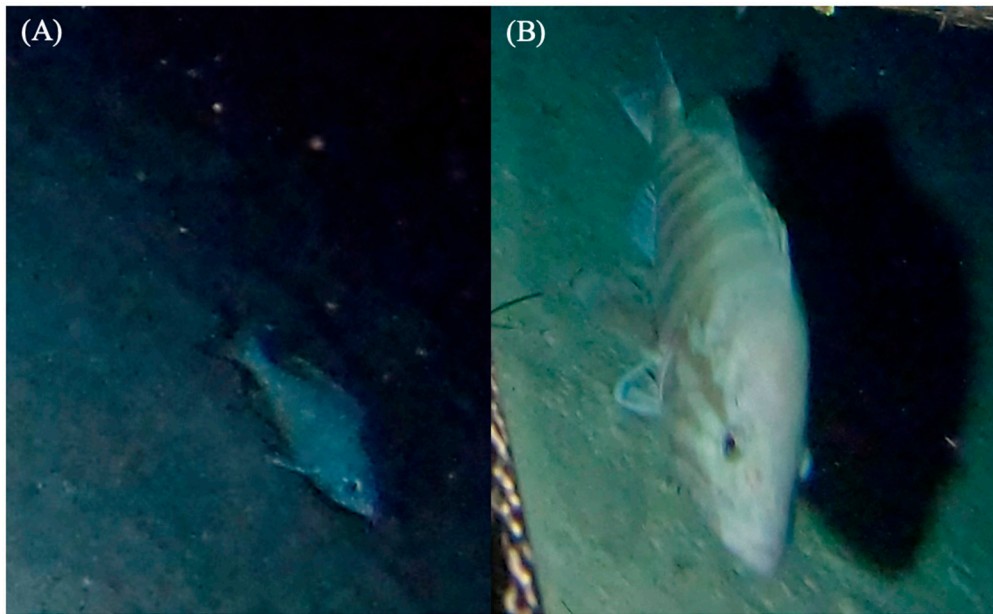

**Figure 3.** In-situ images of Misty Groupers (*Hyporthodus mystacinus*) collected using dBRUV units. (**A**) Observed off the southeast side of the Berry Islands, Bahamas, on 15 January 2020. (**B**) Observed off the southwest side of Grand Cayman, Cayman Islands, on 7 April 2023.

On 7 April 2023, a dBRUV was deployed approximately 1.5 km off the southwest point of Grand Cayman (19.27248° N, −81.40376° W) at a depth of 378 m and recorded 2 h and 34 min of video during the day between 11:10 and 14:03 (Figure 2B). A Misty Grouper, identified using the same morphological criteria as above, was first observed 2 h after deployment, swimming into the detection window and then lingering at the base of the dBRUV unit for approximately 2 s before swimming out of view (Figure 3B) (Supplementary Video S2). During the second appearance, the fish was actively foraging, observing, and hovering over a crustacean that was walking on the benthos before taking the crustacean into its mouth at 29 s in view then spitting out the crustacean at 32 s, and finally, quickly swimming out of view. This appearance totaled 39 s in the detection window. During the third appearance, the fish appeared directly next to the dBRUV on the left-hand side and simply swam across the detection window for a total of 6 s in view. A total of seven appearances were observed during the drop, with behaviors that included slow swimming, hovering, foraging, and looking at something close to the base of the dBRUV unit (based on the vertical body positioning with caudal fin up) for a total time in the detection window of 187 s, and only one individual Misty Grouper being observed at a time. It could not be confirmed whether all grouper detections were the same individual. The habitat observed for this drop site consisted primarily of fine sand (0.063–0.25 mm) with a near-horizontal slope and very low rugosity. There were some anemones observed in the area. In both cases (Bahamas and Cayman), species identification was validated and confirmed by numerous external experts.

Previously, the maximum recorded depth in the primary scientific literature for Misty Grouper was 272.1 m [10], rendering both empirical records from the present study significant depth extensions for the species, ultimately to a new maximum depth of 470 m, with a recorded temperature of 15.79 °C. Considering that the two drop site locations where Misty Groupers were observed in this study were 775 km apart and that the previous scientifically documented maximum depths were similar to each other [9,10], it can be hypothesized that the new depth range of 470 m is likely consistent throughout the species presence in the Caribbean. An observation worth noting is that Misty Groupers have historically been caught along walls and shelves, but both individuals observed during this study were seen on sandy bottom habitats with little slope.

Understanding the connectivity between shallow and deep waters will be a critical consideration in many scientific studies over the next decade, particularly when considering the expectations outlined for the Ocean Decade, which emphasize four main points: capacity development, generating ocean data, building ocean understanding, and increasing use of ocean knowledge [2]. The connection between shallow and deep waters is not only significant when considering nutrient cycling but is also very relevant to animal ecology, oceanography, and sustainable management and conservation measures. When considering ocean nutrient cycles, Misty Groupers aid in nutrient cycling via consumption and excretion as they move vertically throughout the water column, likely playing a role in transporting organic carbon from shallow to deep water [1].

When considering Misty Grouper ecology, it is presumed that they are slow-growing protogynous hermaphrodites, such as is characteristic of epinepheline groupers [5], though there are very few studies that have been conducted for the Misty groupers specifically that can confirm these assumptions [19]. One study observed that the majority of Misty Groupers caught by deep-sea commercial fishermen around the Galapagos Islands were large males, raising and/or reiterating several questions we already had about the species, such as what parameters does the species require for the sex change to occur, do they begin their life in shallow waters and move to the deep-sea once they reach a certain size, and when and where do spawning events occur [12]. Another study revealed that Misty Groupers have a comparatively long lifespan, aging two individuals at approximately 135 and 150 years old [9], meaning that they likely are slower growing than initially anticipated and, correspondingly, may be more heavily affected by anthropogenic impacts compared to faster-growing species that are more biologically productive. Having a better

understanding of the species biology, ecology, and sensitivities is necessary to understand what special considerations they may need when it comes to protection and management and how they may play an important role in developing our understanding of the deep-sea.

Considering how effective small-scale Marine Protected Areas (MPAs) have been thus far [20], the need to establish large-scale MPAs in particularly sensitive and important areas, such as the Greater Caribbean, is a crucial next step towards ocean conservation [3]. These data highlight the potential conservation value in extending existing coastal MPAs into adjacent deep-waters. While this paper reports new information that may improve our understanding, there is little known about the biology and ecology of the Misty Grouper; therefore, they are considered to be Least Concern on the IUCN Red List [19]. Yet, this lack of information puts their sustainability at risk and limits the potential to effectively manage regional stocks of the species throughout its range [21]. Frequent evaluation of economically important fishes is critical for understanding how they are being impacted, both positively and negatively [22,23], and advances in technology have the potential to significantly impact fishery practices [24]. Thus, precautionary protections [25], in the form of expanded coastal MPAs that overlap with the vertical depth range of vulnerable grouper species, may be an effective approach to conservation.

## 4. Conclusions

This report provides critical information towards bettering our understanding of the habitat extent and potentially the ecological role that the Misty Grouper may have. We recognize the dBRUV techniques used here also have limitations, such as the restricted temporal nature of their deployment and their use of lights to illuminate the seafloor for analysis; however, they offer clear benefits for rapidly expanding our potential to survey life on the deep seafloor. Utilizing dBRUV landers for further research surrounding economically and biologically important species will not only allow us to continue to understand the connections throughout the deep-sea and the shallower waters but will also allow for the establishment of better practices and management of marine species. These deep-sea research programs are critical at this pivotal point in the health of the world's oceans, and continued efforts are sure to yield exciting and necessary new discoveries.

**Supplementary Materials:** The following supporting information can be downloaded at: Video S1. Bahamas drop with Misty Grouper appearances: https://www.youtube.com/playlist?list=PLAhaITUgaqubiNSlYnOehDzV7E4uwadst (accessed on 12 March 2024) & Video S2. Cayman drop with Misty Grouper appearances: https://www.youtube.com/playlist?list=PLAhaITUgaquYxpgv-fPbhpzdYIyFUqH_i (accessed on 12 March 2024).

**Author Contributions:** A.J.G., T.F.F., T.A. and M.A.G. led and administered the study. A.J.G., R.F.O., T.A., and M.A.G. conceived the study. S.E.A., O.F.L.D., C.d.S., J.K.K., O.N.S., B.T.P., T.F.F., T.A., R.F.O., M.A.G. and A.J.G. performed the field work. S.E.A. analyzed the data. S.E.A. wrote the first draft of the paper. A.J.G., O.F.L.D., T.F.F., M.A.G., O.N.S., J.K.K., C.d.S. and R.F.O. provided edits on the paper. All authors have read and agreed to the published version of the manuscript.

**Funding:** This work was funded by a Darwin Plus Grant (DPLUS140) from the Department of Environment, Food and Agriculture (DEFRA) of the UK Government and support from the Department of Environment (DOE) of the Cayman Island Government (Cayman field work), and by private philanthropic donations provided to Beneath the Waves (Bahamas field work).

**Institutional Review Board Statement:** The animal study was reviewed and approved by Heriot-Watt University. A certificate of approval was not relevant to this study because there was not any sampling of live animals. All research was covered under permits from the Bahamas Department of Marine Resources, Government of The Commonwealth of The Bahamas, and Department of Environment, Cayman Islands Government.

**Data Availability Statement:** The raw data supporting the conclusions of this article will be made available by the authors without undue reservation.

**Acknowledgments:** We thank the Department of Marine Resources, Bahamas, for the permit and the staff from The Department of Environment Cayman Islands for the permit and their assistance with field work. For assistance with validating species identification, we thank members from the IUCN SSC Groupers and Wrasses Specialist group, particularly M. Craig, C. Malinowski, and B. Ferreira.

**Conflicts of Interest:** The authors declare no conflicts of interest.

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
