# Peer review of "Depth Range Extension for the Misty Grouper Hyporthodus mystacinus Documented via Deep-Sea Landers throughout the Greater Caribbean"

_fishes, doi:10.3390/fishes9040114_

Round 1

Reviewer 1 Report

Comments and Suggestions for Authors

In this contribution, Aldridge et al. note the occurrence of misty groupers at two locations in the Caribbean Sea at depth significantly greater than the previously known depth range of the species. This note is worthy of publication after minor revisions and editing.

The common name of the fish is given throughout the manuscript as “Misty grouper”. It should be either “Misty Grouper” or “misty grouper”. The former if the authors follow AFS naming guidelines, or the latter if they choose not to do so.

Line 41: “etc.” is lazy writing. List additional examples or remove this.

Lines 63–67: This sentence is awkwardly constructed (using both semicolon and colon. I suggest breaking it up into three sentences.

Lines 78–79: “The Bahamas” “archipelago” Also, avoid the use of the semicolon here, or at least use it correctly.

Lines 84–87: This is a circular argument.

Lines 91–92: Avoid usage of semicolons and colons; or at least use them correctly. Typically, they are unnecessary.

Line 96: Opportunities that Caymans present? Rephrase this.

Line 100: Islands

Lines 103–116: This text is verbatim in Gallagher et al. 2023. Rephrase to avoid self-plagiarism or just summarize and reference Gallagher et al. 2023.

Results and Discussion

I suggest providing links to video of all appearance of misty groupers reported in the text. These could be archived on Zenodo or similar, and available from YouTube.

Line 143: “a Cuban dogfish’s” This construction is awkward.

MaxN is used twice as a term but never defined. Define it and cite a source. Since it is MaxN=1 both times it might just be better to avoid the term and state as “only a single individual misty grouper was seen” in any still frame of video.

Line 158: “on the benthic” Benthos?

Line 209: that readily reproduce

Line 225: These depth data...

References in text are given parenthetically with standard syntax (Author1, 2000; Author1 and Author2, 2023, Author3 et al. 1950), but the author instructions stipulate a numbered bibliographic and citation style (https://www.mdpi.com/journal/fishes/instructions#preparation). All entries should be fixed. Additionally, the Literature Cited is rife with inconsistencies and mistakes too numerous (and obvious) to be a reviewer’s responsibility.

Comments on the Quality of English Language

See above

Reviewer 2 Report

Comments and Suggestions for Authors

The depth record of any species is of great biological and ecological significance, and should be compared to the other species in the genus or other species in that ecosystem. Discussions of MPA’s seems unsupported. The focus of this observation on this groupers would benefit from a further discussion of the other members of the genus and their depth ranges.

Given that this genus occurs worldwide and are fisheries target the authors should entertain discussion of this finding upon what is known about deep slope fisheries in the Greater Caribbean. There are at least 4 species of Hyporthodus in the Atlantic Ocean and more worldwide that are part of deep-slope fisheries that could be included in the discussion.

It would be pertinent to include observations of any benthic habitat characteristics observed since this is a critical gap in what is known about their ecology. A more detailed description of the geomorphological characteristics of sites in which these observation were made is needed. The application of this methodology to observe deep slope fishes and their habitat holds much promise, however not much was mentioned about this.

The map figure should include additional bathymetric information to the detail and extent possible, multiple lines of depth contours that provide the slope context or depth near the islands would greatly enhance the ecological value of this note. The maps could be improved as the colored shading is not enough to illustrate the shelf steepness, distance from land or shelf break and orientation.

L15 Failure to define the depths of the ‘deep-sea’ leads to confusion, and this species is not the ’only’ deep slope grouper known. Additional information of the depth records of members of this genus was overlooked and should be included.

L 30-32 Requires a reference. The principal focus of this note is the depth record, not the connectivity per se, perhaps rephrase or limit the text to the depth throughout this .

L 61-62 Requires a reference.

L 75 The fortuitous observation of the grouper at depth answers the question about the maximum depth not the range of depths. Consider rephrasing.

L 80 instead of ‘deep-sea species’ consider the potential for protection of ‘deep-sea habitats’ instead.

L 113 What was the recording interval or if the recording was continuous, please state it.

L 116 What was the wavelength/color of the LED lights? Some species may be attracted or deterred at these depths form different color lights.

L 185 The temperature at depth is key to deep-slope habitats and would benefit from additional oceanographic context for the temperature regimes of the Caribbean deep-water habitats.

L 218-228 This discussion is not really supported by the observation of a specie’s depth record. Consider eliminating. If this is a fisheries target, then the information of their habitat range is what should be discussed, not necessarily a MPA or conservation status.

Adding a clip of the two videos as Supplementary Material would greatly enhance the viewer’s appreciation of the behavior of the species at depth.

This species has been observed at comparable depths by ROV at least during the Okeanos Explorer expedition in 2018. Including this and any other at depth observations would be a meaningful reference to include.

Reviewer 3 Report

Comments and Suggestions for Authors

The minor corrections needed are provided as sticky notes on the pdf.

Suggestions;

Title and Introduction: The name of the author who first defined the species should be given in the title and introduction of the paper.

Introduction Section; Second paragraph; …………..maximum recorded weight of 75.5kg with a total length of 157cm in primary scientific literatüre. Please check!! According to Appeldomm et al. (1997), This species was reported for male as 160 cm for females 100 cm, and it is published max. weight as 107 kg.

Reference; Appeldoorn, R.S., G.D. Dennis and O.M. Lopez, 1987. Review of shared demersal resources of Puerto Rico and the Lesser Antilles region. FAO Fish. Rep. 383:36-104.

Material and Methods; The sardine used as bait is given as Atlantic bonito Sarda sarda. The species of sardine should be written correctly again.

Figure 3; It would be appropriate to present the images taken from the video camera more clearly as pictures.

Best wishes

Comments on the Quality of English Language

The minor corrections needed are provided as sticky notes on the pdf.

Reviewer 4 Report

Comments and Suggestions for Authors

Dear Authors.
I have added comments and suggestions on the attached. 

I think the paper could be greatly reduced without compromising your findings. I think putting a wider (global) context may be useful and I have provided additional papers for your information. Many others exist. 
I hope you find my comments useful to your review

Reviewer 5 Report

Comments and Suggestions for Authors

The understanding of flagship species or top predators in deep-sea ecosystems is very helpful to improve human understanding of the ocean, and is necessary to strengthen the protection of keystone species and ecosystems. In fact, the extreme conditions of the deep sea, such as high pressure, darkness, and difficulty in collecting biological samples, have become huge obstacles to studying deep-sea organisms and ecosystems. The development of new observation methods, such as video acquisition and acoustic sampling, has become a trend in deep-sea biology research. So, I think the content of this manuscript is interesting. Although the content of the manuscript does not seem very rich. The MS “Depth range extension for the Misty grouper Hyporthodus mystacinus documented via deep-sea landers throughout the Greater Caribbean” documents the presence of Misty groupers via deep-sea lander videos at depth up to 470m, extending the previously known depth range for the species. Additionally, the overall study emphasizes the need to better understand deep-sea connectivity, diversity, and importance, in today’s changing oceans.

1.      The manuscript contains less information about the biology of this species. Misty groupers are one of the largest and most geographically widespread grouper species, and one of the only grouper species known to occur in the deep-sea. Overall, groupers are significantly important to both recreational and commercial fisheries throughout the Atlantic, particularly the Misty grouper for deep-sea commercial fisheries. It is amazing that such an important species, both ecologically and commercially, has received so little attention and even less research.

2.      What is the detection range of the camera of the deep-sea BRUV (dBRUV)?

3.      What does the Misty grouper feed on?  Why are (Sarda sarda) used as bait?

4.      The manuscript described that the study was based on multiple years of observational data, but the manuscript did not seem to explain what years of data were available, and the observational data were not that rich.

5.      In lines 185-186: Considering that the two drop sited locations were 775km apart, and that the previous scientifically documented maximum depths were similar to each other, it can be assumed that the depth range is likely consistent throughout the species presence in the Caribbean. Does the author mean that 470m is similar to 272m?

6.      About the suggestion small-scale Marine Protected Areas (MPAs), I think it is a little out of place, because many aspects of Misty grouper are not clear, such as the amount of resources? Development status? Overfishing? So, I think, this suggestion is not very relevant to the manuscript. Perhaps, the authors could envision how to build on the present work next.
